# Immune escape and attenuated severity associated with the SARS-CoV-2 BA.2.86/JN.1 lineage

Joseph A. Lewnard [1] ✉, Parag Mahale[2], Debbie Malden [2], Vennis Hong[2], Bradley K. Ackerson [2], Bruno J. Lewin [2], Ruth Link-Gelles [3], Leora R. Feldstein [4], Marc Lipsitch[4] & Sara Y. Tartof [2,5]

The SARS-CoV-2 BA.2.86 lineage, and its sublineage JN.1 in particular, achieved widespread transmission in the US during winter 2023–24. However, this surge in infections was not accompanied by COVID-19 hospitalizations and mortality commensurate with prior waves. To understand shifts in COVID-19 epidemiology associated with JN.1 emergence, we compared characteristics and clinical outcomes of time-matched cases infected with BA.2.86 lineages (predominantly representing JN.1) versus co-circulating XBB-derived lineages in December, 2023 and January, 2024. Cases infected with BA.2.86 lineages received greater numbers of COVID-19 vaccine doses, including XBB.1.5-targeted boosters, in comparison to cases infected with XBB-derived lineages. Additionally, cases infected with BA.2.86 lineages experienced greater numbers of documented prior SARS-CoV-2 infections. Cases infected with BA.2.86 lineages also experienced lower risk of progression to severe clinical outcomes requiring emergency department consultations or hospital admission. Sensitivity analyses suggested under-ascertainment of prior infections could not explain this apparent attenuation of severity. Our findings implicate escape from immunity acquired from prior vaccination or infection in the emergence of the JN.1 lineage and suggest infections with this lineage are less likely to experience clinically-severe disease. Monitoring of immune escape and clinical severity in emerging SARS-CoV-2 variants remains a priority to inform responses.

The BA.2.86 SARS-CoV-2 lineage, which is distinguished from the parent BA.2 lineage by over 30 mutations in the spike protein, was detected simultaneously in multiple European countries in July, 2023[1]. A sublineage (BA.2.86.1.1; "JN.1") harboring one additional Spike (S) protein mutation (L455S) emerged shortly thereafter and became the dominant circulating lineage in the US by late December, 2023[2]. Similar to other BA.2.86 lineages, JN.1 has been reported to evade neutralizing antibody responses associated with prior SARS-CoV-2 infection and COVID-19 vaccination in comparison to co-circulating lineages derived from XBB.1.5[3,4]. Modification of angiotensin-converting enzyme 2 (ACE2) binding affinity may have further contributed to the establishment of JN.1 and other BA.2.86 lineages[5–7].

[1]School of Public Health, University of California, Berkeley, Berkeley, CA, USA. [2]Department of Research & Evaluation, Kaiser Permanente Southern California, Pasadena, CA, USA. [3]Coronavirus and Other Respiratory Viruses Division, National Center for Immunization and Respiratory Diseases, US Centers for Disease Control & Prevention, Atlanta, GA, USA. [4]COVID-19 Response Team, Centers for Disease Control and Prevention, Atlanta, GA, USA. [5]Department of Health Systems Science, Kaiser Permanente Bernard J. Tyson School of Medicine, Pasadena, CA, USA. ✉e-mail: jLewnard@berkeley.edu

While declining rates of clinical SARS-CoV-2 testing prevent comparison of case-based surveillance of JN.1 with earlier phases of the pandemic, detection of SARS-CoV-2 genetic material in wastewater during the JN.1 wave reached levels not seen since the peak of the Omicron BA.1 wave in January, 2021[8]. However, this expansive transmission of JN.1 was not associated with increases in COVID-19-related hospital admissions or deaths commensurate with the earlier BA.1, BA.4/BA.5, and XBB/XBB.1.5 epidemic waves[9]. Assessments of characteristics and clinical outcomes of cases infected with emerging SARS-CoV-2 lineages are needed to interpret whether such epidemiologic observations reflect changes in clinical severity and immune protection[10–15]. We, therefore, compared prior vaccination, documented SARS-CoV-2 infection history, and post-diagnosis healthcare utilization among cases infected with differing lineages within the Kaiser Permanente Southern California (KPSC) healthcare system who were tested in outpatient settings during December 2023 and January 2024.

## Results

### Study setting, enrollment, and case definitions

The KPSC healthcare system provides managed, integrated care spanning virtual, outpatient, emergency department, and inpatient settings to 4.7 million adults residing in southern California, representing roughly 20% of the region's population. Individuals are enrolled in KPSC plans through employer-sponsored, pre-paid, or government-subsidized coverage schemes. Enrolled members closely resemble the general insured population within Southern California[16,17]. Electronic health care records capture all in-network care delivery, comprising diagnoses, prescription fills, procedures, laboratory testing, vaccinations, and clinical notes. Records of COVID-19 vaccinations received outside KPSC are imported from the California Immunization Registry[18]. Other care delivered out-of-network is ascertained through insurance claim reimbursements, enabling near-complete ascertainment of members' medical histories.

We conducted a retrospective cohort study leveraging the opportunity for longitudinal follow-up among cases initially diagnosed with SARS-CoV-2 infection in outpatient settings to monitor progression to severe disease outcomes. Our analyses followed cases who had been members of KPSC health plans for ≥1 year from the point of their first documented positive outpatient test between 1 December, 2023 and 30 January, 2024. Over this period, 46,067 eligible individuals tested positive for SARS-CoV-2 in KPSC outpatient settings. Of this population, 7694 (17%) had tests processed by regional testing laboratories using the TaqPath COVID-19 Combo Kit assay (Thermo

Fisher Scientific, Waltham, Massachusetts), which provides readout on probes for the S, N, and orf1a/b genes (Fig. 1 and Table S1). Dropout of the S gene probe in samples that tested positive for both N and orf1a/b (defined as cycle threshold [$c_T$] values of ≥37 for S and <37 for N and orf1a/b) provided 98–100% sensitivity and 96% specificity for distinguishing BA.2.86-derived lineages within a sample of 1078 sequenced specimens from KPSC testing laboratories during the study period (Table S2), consistent with observations in other settings[19].

We therefore used S-gene target failure (SGTF) as a proxy for infection with JN.1 or other BA.2.86 lineages and defined the primary analytic cohort as the subset of cases whose specimens were processed using TaqPath COVID-19 Combo Kit assays ($N = 7694$). This population closely resembled other outpatient-diagnosed cases at KSPC over the same period in terms of sex, health status, prior-year healthcare utilization, and community socioeconomic characteristics. However, cases tested via TaqPath COVID-19 Combo Kit assays were modestly younger in comparison to other outpatient-diagnosed cases (median age 46 vs. 50 years, respectively) and more racially and ethnically diverse (22 vs 34% identifying as non-Hispanic White, respectively; Table S1). Within this primary analytic cohort, cases infected with JN.1 or other BA.2.86 lineages ($N = 3080$) did not differ appreciably from those infected with non-BA.2.86 lineages ($N = 4614$) in terms of age, sex, or racial/ethnic distribution, comorbidity burden, prior-year patterns of healthcare utilization, community socioeconomic characteristics, or receipt of nirmatrelvir-ritonavir (Table 1).

### Comparison of immune history by infecting lineage

We compared vaccination history among cases infected with BA.2.86 lineages or non-BA.2.86 lineages, hypothesizing that immune escape by BA.2.86 would lead to the detection of related lineages among individuals with a history of COVID-19 vaccination. Because an updated XBB.1.5-targeted monovalent vaccine was the primary public health strategy for preventing COVID-19 during the study period, we first compared receipt of this vaccine among cases infected with BA.2.86 lineages and non-BA.2.86 lineages (Table 2). Overall, 16% (501/3,080) of cases infected with BA.2.86 lineages and 12% (573/4,614) of cases infected with other lineages received an XBB.1.5-targeted monovalent vaccine. Via conditional logistic regression analyses matched on the testing week and controlling for measured characteristics of cases (see Methods), we estimated that cases infected with BA.2.86 lineages had 14% (95% confidence interval: 1–28%) higher adjusted odds of XBB.1.5-targeted monovalent vaccination in comparison to cases infected with other lineages. Cases infected with BA.2.86 lineages also had 10% (1–20%) higher adjusted odds of having received BA.4./BA.5-targeted

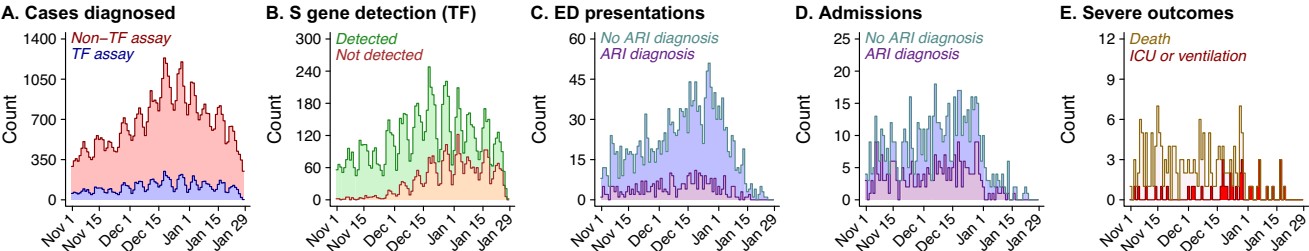

**Fig. 1 | Testing, S-gene targeted taction, and clinical outcomes during the study period.** Panels illustrate **A** the number of outpatient cases diagnosed daily from tests processed on Thermo Fisher TaqPath COVID-19 Combo Kit (TF) assays or non-TF assays; **B** the daily frequency of TF-tested specimens yielding positive results with S gene detected (non-BA.2.86 lineages) or S-gene target failure (BA.2.86-derived lineages); **C** the daily frequency of outpatient cases with positive SARS-CoV-2 testing results (organized by date of test) who experienced emergency department (ED) presentations within 14 days of testing, stratified according to presence or absence of acute respiratory infection (ARI) diagnoses associated with their ED presentation; **D** the daily frequency of outpatient cases with positive SARS-CoV-2

testing results (organized by date of test) who experienced hospital admission within 28 days of testing, stratified according to presence or absence of ARI diagnoses associated with their hospital admission; and **E** the daily frequency of outpatient cases with positive SARS-CoV-2 testing results (organized by date of test) who experienced intensive care unit (ICU) admission, initiation of mechanical ventilation, or death within 60 days of testing. Analyses include data from 46,067 eligible individuals throughout the study period, including 7,694 with TF-tested specimens. Source data to replicate the figure are provided as a Source Data file (fig1_source.xlsx).

**Table 1 | Characteristics of individuals infected with SARS-CoV-2 according to lineage**

| Characteristic | Cases, n/N (%) | |
| --- | --- | --- |
| | S-gene detected (non-BA.2.86 lineage) | S-gene target failure (BA.2.86-derived lineage) |
| | N = 4614 | N = 3080 |
| Age (years)[a] | | |
| 0–9 | 228 (4.9) | 95 (3.1) |
| 10–19 | 258 (5.6) | 129 (4.2) |
| 20–29 | 423 (9.2) | 314 (10.2) |
| 30–39 | 713 (15.5) | 560 (18.2) |
| 40–49 | 778 (16.9) | 646 (21.0) |
| 50–59 | 785 (17.0) | 574 (18.6) |
| 60–69 | 708 (15.3) | 409 (13.3) |
| 70–79 | 469 (10.2) | 229 (7.4) |
| ≥80 | 252 (5.5) | 124 (4.0) |
| Sex | | |
| Female | 2800 (60.7) | 1848 (60.0) |
| Male | 1814 (39.3) | 1232 (40.0) |
| Race/ethnicity | | |
| White, non-Hispanic | 1076 (23.3) | 593 (19.3) |
| Black, non-Hispanic | 459 (9.9) | 344 (11.2) |
| Hispanic (any race) | 2189 (47.4) | 1472 (47.8) |
| Asian | 570 (12.4) | 444 (14.4) |
| Pacific Islander | 41 (0.9) | 34 (1.1) |
| Other/mixed/ unknown race | 279 (6.3) | 193 (6.3) |
| Body mass index[a] | | |
| Underweight (<18.5) | 205 (4.4) | 75 (2.4) |
| Normal weight (18.5–24.9) | 958 (20.8) | 625 (20.3) |
| Overweight (25.0–29.9) | 1255 (27.2) | 822 (26.7) |
| Obese (≥30.0) | 1715 (37.2) | 1192 (38.7) |
| Cigarette smoking[a] | | |
| Never smoker | 3263 (70.7) | 2132 (69.2) |
| Former smoker | 838 (18.2) | 528 (17.1) |
| Current smoker | 170 (3.7) | 113 (3.7) |
| Charlson comorbidity index | | |
| 0 | 2756 (59.7) | 1984 (64.4) |
| 1–2 | 1273 (27.6) | 825 (26.8) |
| 3–5 | 415 (9.0) | 201 (6.5) |
| ≥6 | 170 (3.7) | 70 (2.3) |
| Prior-year healthcare utilization | | |
| 0–9 outpatient encounters | 2365 (51.3) | 1614 (52.4) |
| 10–19 outpatient encounters | 1213 (26.3) | 778 (25.3) |
| 20–29 outpatient encounters | 519 (11.2) | 374 (12.1) |
| ≥30 outpatient encounters | 517 (11.2) | 314 (10.2) |
| Any emergency department presentation | 1039 (22.5) | 643 (20.9) |
| Any inpatient admission | 278 (6.0) | 140 (4.5) |
| Census tract median household income[a] | | |
| <$40,000 | 209 (4.5) | 138 (4.5) |
| $40,000-79,999 | 1808 (39.2) | 1246 (40.5) |
| $80,000-119,999 | 1537 (33.3) | 1042 (33.8) |

**Table 1 (continued) | Characteristics of individuals infected with SARS-CoV-2 according to lineage**

| Characteristic | Cases, n/N (%) | |
| --- | --- | --- |
| | S-gene detected (non-BA.2.86 lineage) | S-gene target failure (BA.2.86-derived lineage) |
| $120,000-159,999 | 657 (14.2) | 423 (13.7) |
| ≥$160,000 | 224 (4.9) | 137 (4.4) |
| Receipt of nirmatrelvir-ritonavir | | |
| Received within ≤5 days from diagnosis | 909 (19.7) | 546 (17.7) |
| Received >5 days from diagnosis | 8 (0.2) | 2 (0.1) |
| Not received | 3697 (80.1) | 2532 (82.2) |

Data encompass the primary analytic cohort, comprised of individuals testing positive for SARS-CoV-2 from tests undertaken in outpatient settings between 1 December, 2023 and 30 January, 2024 which were processed via TaqPath COVID-19 Combo Kit assays, who belonged to KPSC health plans for at least one year prior to their index test date. Characteristics of all eligible cases and those diagnosed on tests processed via TaqPath COVID-19 Combo Kit assays are presented in Table S1.
[a]Counts and percentages are counted excluding missing values (9630 for body mass index; 8535 for cigarette smoking; 4129 for census tract median household income; and 1102 for age.

bivalent COVID-19 vaccine doses, and 28% (13–45%) higher adjusted odds of having received both Omicron-adapted vaccine products.

Due to the low uptake of updated COVID-19 vaccine formulations within the study population, we also assessed the total number of COVID-19 vaccine doses received among cases infected with BA.2.86 lineages and non-BA.2.86 lineages. Adjusted odds of receipt of 5, 6, and ≥7 COVID-19 vaccine doses were 38% (95% confidence interval: 9–74%), 51% (17-95%), and 60% (7–138%) higher among cases infected with BA.2.86 lineages in comparison to cases infected with non-BA.2.86 lineages (Table 2). Cases infected with BA.2.86 lineages also had 4–20% higher adjusted odds of having received 1–4 COVID-19 vaccine doses, although the possibility of no difference could not be excluded within some smaller case strata for these lower-dose exposures. Similar patterns persisted in subgroup analyses restricted to cases documented to have experienced ≥1 or ≥2 prior SARS-CoV-2 infections, suggesting relationships between prior vaccination and infecting lineage were not exclusively mediated by antecedent effects of vaccination on cases' risk of prior infection (Table S3). Following adjustment for the number and type of vaccine doses received, differences in the timing of cases' most recent COVID-19 vaccine doses were not apparent between cases infected with BA.2.86 lineages or non-BA.2.86 lineages.

We next compared the history of documented prior SARS-CoV-2 infection among cases infected with BA.2.86 lineages and non-BA.2.86 lineages. Among cases infected with BA.2.86 lineages and non-BA.2.86 lineages, 54 and 49%, respectively, had no documented history of SARS-CoV-2 infection (Table 2); adjusted odds of any documented prior infection were 9% (2–18%) higher among cases infected with BA.2.86 lineages than among cases infected with non-BA.2.86 lineages. Point estimates of the association of infecting lineage with cases' number of documented prior infections were consistent with a dose-response relationship, although statistical precision was limited, and under-ascertainment of prior infections could lead to under-estimation of effect sizes for associations. Adjusted odds of 1, 2, and ≥3 prior documented infections were 8% (0–17%), 13% (−1–29%), and 30% (−11–91%) higher among cases infected with BA.2.86 lineages than among cases infected with non-BA.2.86 lineages.

In analyses distinguishing the periods during which cases' prior infections occurred, documented infection during the period when XBB lineages were dominant in circulation (1 December, 2022 to 31 October, 2023) was more common among cases infected with BA.2.86 lineages than among cases infected with non-BA.2.86 lineages (adjusted odds ratio = 1.16 [1.02–1.32]; Table S4). However, documented

**Table 2 | Prior vaccination and documented SARS-CoV-2 infection among individuals infected with SARS-CoV-2 according to infecting lineage**

| Exposure | Cases, n/N (%) | | Odds ratio (95% CI), JN.1 vs. non-JN.1 infection | |
|---|---|---|---|---|
| | S-gene detected (non-BA.2.86 lineage) | S-gene target failure (BA.2.86-derived lineage) | Unadjusted[a] | Adjusted[b] |
| | N = 4614 | N = 3080 | | |
| Receipt of updated COVID-19 vaccines[c] | | | | |
| No XBB.1.5 (monovalent) vaccine doses | 4041 (87.6) | 2579 (83.7) | ref. | ref. |
| Any XBB.1.5 (monovalent) vaccine doses | 573 (12.4) | 501 (16.3) | 1.23 (1.11, 1.35) | 1.14 (1.01, 1.28) |
| No BA.4/BA.5 (bivalent) vaccine doses | 3205 (69.5) | 1996 (64.8) | ref. | ref. |
| Any BA.4/BA.5 (bivalent) vaccine doses | 1409 (30.5) | 1084 (35.2) | 1.14 (1.06, 1.23) | 1.10 (1.01, 1.20) |
| 0 Omicron-targeted vaccine doses | 3096 (67.1) | 1927 (62.6) | ref. | ref. |
| Any Omicron-targeted vaccine | 1518 (32.9) | 1153 (37.4) | 1.13 (1.05, 1.22) | 1.12 (1.03, 1.21) |
| Both BA.4/BA.5 (bivalent) and XBB1.5 (monovalent) vaccines | 464 (10.1) | 432 (14.0) | 1.43 (1.15, 1.43) | 1.28 (1.13, 1.45) |
| Number of vaccine doses received | | | | |
| 0 vaccine doses | 569 (12.3) | 272 (8.8) | ref. | ref. |
| 1 vaccine dose | 122 (2.6) | 66 (2.1) | 1.08 (0.83, 1.41) | 1.05 (0.80, 1.38) |
| 2 vaccine doses | 901 (19.5) | 564 (18.3) | 1.16 (1.01, 1.34) | 1.10 (0.94, 1.28) |
| 3 vaccine doses | 1453 (31.5) | 1011 (32.8) | 1.27 (1.11, 1.45) | 1.20 (1.04, 1.39) |
| 4 vaccine doses | 824 (17.9) | 570 (18.5) | 1.25 (1.08, 1.44) | 1.23 (1.05, 1.44) |
| 5 vaccine doses | 415 (9.0) | 331 (10.7) | 1.34 (1.14, 1.58) | 1.43 (1.20, 1.71) |
| 6 vaccine doses | 285 (6.2) | 231 (7.5) | 1.32 (1.11, 1.58) | 1.57 (1.28, 1.91) |
| ≥7 vaccine doses | 45 (1.0) | 35 (1.1) | 1.32 (0.93, 1.88) | 1.69 (1.16, 2.45) |
| Timing of prior vaccination[c] | | | | |
| No doses received | 569 (12.3) | 272 (8.8) | ref. | ref. |
| Last vaccine dose within <3 months | 527 (11.4) | 436 (14.2) | 1.18 (1.07, 1.31) | 1.05 (0.80, 1.38) |
| Last vaccine dose within 3–6 months | 79 (1.7) | 92 (3.0) | 1.35 (1.10, 1.66) | 1.09 (0.79, 1.52) |
| Last vaccine dose >6 months prior | 3439 (74.5) | 2280 (74.0) | 0.98 (0.90, 1.06) | 0.97 (0.81, 1.17) |
| Documented prior infection | | | | |
| 0 documented infections | 2505 (54.3) | 1506 (48.9) | ref. | ref. |
| Any prior infection | 2109 (45.7) | 1574 (51.1) | 1.14 (1.06, 1.22) | 1.09 (1.02, 1.18) |
| 1 documented infection | 1753 (38.0) | 1269 (41.2) | 1.11 (1.03, 1.20) | 1.08 (1.00, 1.17) |
| 2 documented infections | 332 (7.2) | 278 (9.0) | 1.17 (1.03, 1.33) | 1.14 (0.99, 1.30) |
| ≥3 documented infections | 24 (0.5) | 27 (0.9) | 1.37 (0.93, 2.00) | 1.30 (0.89, 1.91) |

Data encompass the primary analytic cohort (N = 7694 individuals), comprised of individuals testing positive for SARS-CoV-2 from tests undertaken in outpatient settings between 1 December, 2023 and 30 January, 2024 which were processed via TaqPath COVID-19 Combo Kit assays, who belonged to KPSC health plans for at least 1 year prior to their index test date.
[a]Unadjusted odds ratios are computed via conditional logistic regression models matching on week of testing alone.
[b]Adjusted odds ratios are computed via conditional logistic regression models matching on week of testing and controlling for age, sex, race/ethnicity, body mass index, history of cigarette smoking, prior-year healthcare utilization across all settings, Charlson comorbidity index, and median household income within cases' census tract according to the categorization scheme indicated in Table 1. Missing values were addressed via multiple imputation, with results pooled across five pseudo-dataset replicates.
[c]Analyses of vaccine type and timing adjust for a number of monovalent wild-type (Wuhan-Hu-1) vaccine doses received.

prior infection during the period when BA.2 lineages were dominant (3 February to 24 June, 2022) was also more common among cases infected with BA.2.86 lineages than among cases infected with non-BA.2.86 lineages (adjusted odds ratio = 1.16 [1.02–1.32]), suggesting that responses to infection with ancestral BA.2 lineages did not confer greater protection against BA.2.86 lineages in comparison to co-circulating XBB-derived lineages.

**Comparison of immune history by calendar period**

To overcome limitations in statistical power affecting comparisons within the primary analytic cohort, we also assessed outcomes among cases diagnosed in all outpatient settings over bimonthly intervals throughout the study period. Although determination of individual-level lineage was not possible for cases tested on assays without SGTF results, we expected differences in immune history among cases infected with BA.2.86 lineages and non-BA.2.86 lineages would be reflected among cases diagnosed at differing points in time during the

expansion of the JN.1 lineage. Consistent with our primary results obtained at the level of individual infection genotype, cases diagnosed at later points in time (i.e., as BA.2.86 lineages became dominant in circulation) tended to have received greater numbers of COVID-19 vaccine doses, and to have experienced greater numbers of prior documented SARS-CoV-2 infections (through the period ending 31 October, 2023) in comparison to cases diagnosed in November, 2023 (Fig. 2 and Table S5). Compared to those diagnosed in November, 2023, cases diagnosed between 16-30 January, 2024 had 18% (4–34%) and 39% (14–70%) higher odds of having received 6 and ≥7 COVID-19 vaccine doses, respectively. Similarly, adjusted odds of 1, 2, and ≥3 prior documented SARS-CoV-2 infections were 13% (8–19%), 18% (7–29%), and 28% (−1–67%) higher among cases diagnosed between 16 and 30 January, 2024 in comparison to those diagnosed between 1 and 30 November, 2023. Implementation of XBB-targeted monovalent COVID-19 vaccines throughout Fall, 2023 prevented similar period-based comparisons for receipt of updated vaccines.

### A. Association of period with vaccine doses received

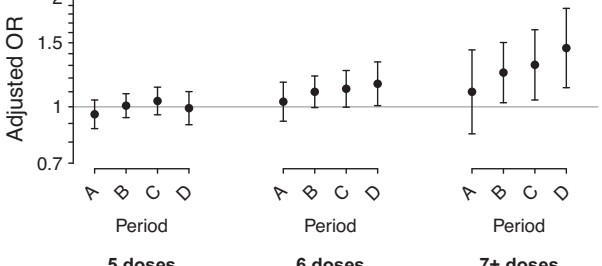

### B. Association of period with documented prior infection

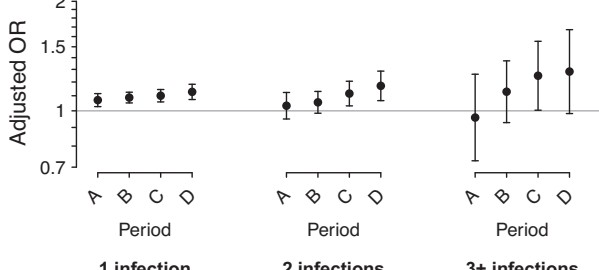

### C. Association with progression risk

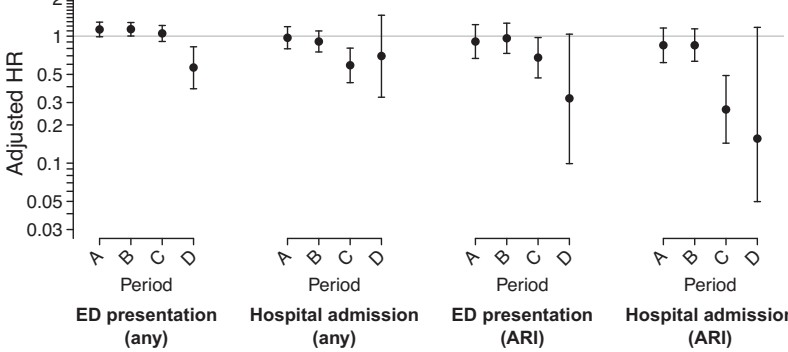

**Fig. 2 | Period-based comparison of prior vaccination, prior documented infection, and risk of progression to various clinical outcomes.** Panels illustrate **A** adjusted odds ratios, fitted via logistic regression models, for receipt of 5, 6, or ≥7 COVID-19 vaccine doses (relative to zero doses) among all outpatient cases diagnosed in the indicated periods relative to those diagnosed between 1 and 30 November, 2023; **B** adjusted odds ratios, fitted via logistic regression models, for documentation of 1, 2, or ≥3 prior SARS-CoV-2 infections (relative to zero documented prior SARS-CoV-2 infections) among all outpatient cases diagnosed in the indicated periods relative to those diagnosed between 1 and 30 November, 2023; and **C** adjusted hazard ratios, fitted via Cox proportional hazards models, for progression to emergency department (ED) presentation or hospital admission, due to any cause or in association with acute respiratory infection (ARI) diagnoses, comparing outpatient cases diagnosed in the indicated periods to those diagnosed between 1 and 30 November, 2023. All models adjust for age, sex, race/ethnicity,

body mass index, history of cigarette smoking, prior-year healthcare utilization across all settings, Charlson comorbidity index, and median household income within cases' census tract according to the categorization scheme indicated in Table 1. In addition, Cox proportional hazards models adjust for nirmatrelvir-ritonavir receipt as a time-varying exposure. Missing values were addressed via multiple imputation, with results pooled across five pseudo-dataset replicates. Tests for non-zero slopes in the Schoenfeld residuals of Cox proportional hazards models identified no violations of the proportional hazards assumption (two-sided $p > 0.1$ for all fitted models). Analyses include data from 46,067 eligible individuals throughout the study period. For all panels (**a**–**c**), points indicate maximum likelihood estimates, with surrounding lines delineating 95% confidence intervals; we generated estimates via Cox proportional hazards models (**A**, **B**) and conditional logistic regression models (**c**). Source data to replicate the figure are provided as a Source Data file (fig2_source.xlsx).

## Comparison of clinical outcomes by infecting lineage

We next assessed the risk of clinical outcomes signifying disease progression following an initial outpatient diagnosis within our primary analytic cohort. Outcomes of interest included emergency department presentation within 14 days, hospital admission within 28 days, and intensive care unit (ICU) admission, mechanical ventilation, or death within 60 days. Rates of these outcomes were 24.6, 4.0, and 0.8 events, respectively, per 10,000 person-days of follow-up among cases infected with non-BA.2.86 lineages and 11.4, 1.6, and 0.4 events, respectively, per 10,000 person-days among cases infected with BA.2.86 lineages (Table 3). In Cox proportional hazards models matching cases on the testing week and controlling for measured case characteristics, including vaccination and documented prior infection (see Methods), adjusted hazards of emergency department presentation and hospital admission were 54% (32–69%) and 51% (−15–79%) lower, respectively, among cases infected with BA.2.86 lineages than among cases infected with non-BA.2.86 lineages. Adjustment for risk factors was not feasible for analyses of ICU admission, mechanical ventilation, or death due to the low frequency of such outcomes. Expecting that some emergency department presentations and hospital admissions following outpatient SARS-CoV-2 detections would be attributable to factors unrelated to COVID-19, we also conducted analyses restricting outcomes to emergency department

presentations or hospital admissions associated with acute respiratory infection (ARI) diagnosis codes (Table S6). Adjusted hazards of ARI-associated emergency department presentations and hospital admissions were 62% (−2–86%) and 85% (−12–98%) lower, respectively, among cases infected with BA.2.86 lineages than among cases infected with non-BA.2.86 lineages (Table 3).

## Comparison of clinical outcomes by calendar period

While the above findings were consistent with a scenario in which BA.2.86 lineages were associated with attenuated clinical severity, comparisons for most outcomes lacked sufficient statistical power to exclude the possibility of no difference. To overcome this limitation, we next compared outcomes among cases diagnosed during successive bimonthly periods throughout the study period as JN.1 became the dominant circulating variant (Fig. 2). Compared to cases tested between 1 and 30 November, 2023, adjusted hazard ratios for emergency department presentation were 1.05 (0.91–1.21) for cases tested between 1 and 15 January, 2024 and 0.57 (0.39–0.82) for cases tested between 16 and 30 January, 2024. For the outcome of ARI-associated emergency department presentations, adjusted hazard ratios declined to 0.68 (0.47–0.97) and 0.32 (0.10–1.04), respectively, for cases tested between 1 and 15 January and 16 and 30 January, 2024. For the same periods, adjusted hazard ratios of hospital admission were 0.59

**Table 3 | Clinical progression among individuals infected with SARS-CoV-2 according to infecting lineage**

| Episode type | Outcome | Events, n (Rate per 10,000 days) | | Hazard ratio (95% CI), JN.1 vs. non-JN.1 infection | |
|---|---|---|---|---|---|
| | | S-gene detected (non-BA.2.86 lineage) | S-gene target failure (BA.2.86-derived lineage) | Unadjusted[a] | Adjusted[b] |
| | | N = 4614 | N = 3080 | | |
| Episodes associated with all causes | | | | | |
| | Emergency department presentation | 126 (24.6) | 33 (11.4) | 0.41 (0.27, 0.60) | 0.47 (0.31, 0.70) |
| | Hospital admission | 36 (4.0) | 7 (1.6) | 0.33 (0.15, 0.76) | 0.50 (0.22, 1.13) |
| | ICU admission, mechanical ventilation, or death | 9 (0.8) | 2 (0.4) | 0.40 (0.08, 1.89) | – – |
| | Death | 5 (0.5) | 1 (0.2) | 0.38 (0.04, 3.33) | – – |
| ARI-associated episodes[c] | | | | | |
| | Emergency department presentation | 22 (4.2) | 5 (1.7) | 0.35 (0.13, 0.96) | 0.40 (0.14, 1.13) |
| | Hospital admission | 18 (2.0) | 1 (0.2) | 0.09 (0.01, 0.68) | 0.13 (0.02, 1.04) |

Data encompass the primary analytic cohort (N = 7694 individuals), comprised of individuals testing positive for SARS-CoV-2 from tests undertaken in outpatient settings between 1 December, 2023 and 30 January, 2024 which were processed via TaqPath COVID-19 Combo Kit assays, who belonged to KPSC health plans for at least 1 year prior to their index test date. We verified that the proportional hazards assumption was met by visual examination of parallel trends in Kaplan–Meier curves (Fig. S1) and by testing for non-zero slopes of Schoenfeld residuals from fitted models; for each fitted model, this test yielded two-sided $p > 0.1$.
[a]Unadjusted hazard ratios are computed via Cox proportional hazards regression models matching on week of testing alone.
[b]Adjusted hazard ratios are computed via Cox proportional hazards regression models matching on the week of testing and controlling for age, sex, race/ethnicity, body mass index, history of cigarette smoking, prior-year healthcare utilization across all settings, Charlson comorbidity index, and median household income within cases' census tract according to the categorization scheme indicated in Table 1. In addition, nirmatrelvir-ritonavir receipt is defined as a time-varying exposure. Missing values were addressed via multiple imputation, with results pooled across five pseudo-dataset replicates.
[c]Acute respiratory infection diagnosis codes are presented in Table S6.

(0.43–0.80) and 0.70 (0.33–1.46), respectively, and adjusted hazard ratios of ARI-associated hospital admission were 0.26 (0.14–0.49) and 0.16 (0.05–1.17), respectively.

In interpreting findings of the period-based analysis, it is important to consider that differences over time in the clinical threshold at which cases sought SARS-CoV-2 testing or subsequently presented for care—especially during the holiday season[20]—could hinder attribution of differences in risk by calendar period to the emergence of the JN.1 lineage. For instance, we observed transient increases in the risk of emergency department presentation and hospital admission or ARI-associated hospital admission among cases diagnosed between December 7–10 and December 24–31, 2023 during the Hannukah and Christmas/New Year holidays, respectively (Fig. 3). Estimated reductions in risk across calendar periods by late January, 2024, exceeded expectations based on our estimates of the difference in risk for each clinical outcome associated with BA.2.86 lineages and the proportion of cases caused by BA.2.86 lineages (Table S5; Methods). Thus, reductions in risk of severe outcomes over the study period could be explained only partially by the expansion of BA.2.86 lineages.

**Sensitivity analyses addressing unobserved prior infections**
Our inability to control completely for cases' infection history further limits our comparison of clinical outcomes according to infecting lineage. In the event that cases infected with BA.2.86 experienced greater numbers of unobserved as well as observed infections in comparison to cases infected with non-BA.2.86 lineages, protection from these unobserved infections could contribute to the apparent association of infecting lineage (or infection during BA.2.86-dominant periods) with attenuated risk of clinical progression[21]. Restricting the sample to cases who had >5 healthcare interactions in the preceding year (N = 5167 members of the primary analytic cohort)—among whom we expected that prior infections would have been recorded with greater likelihood—yielded results confirming those of the primary analyses (Table S7), although this approach was not guaranteed to eliminate potential bias due to unrecorded infections. We, therefore, undertook sensitivity analyses imputing alternative individual-level infection histories to account for the possibility that unobserved prior infections were especially prevalent among individuals infected with

BA.2.86 lineages and those who evaded severe outcomes (see Methods).

Extreme conditions of differential ascertainment of prior infections were needed to arrive at scenarios in which BA.2.86 lineages were not associated with attenuated risk of clinical progression. For both outcomes, the direction of association was reversed (resulting in a statistically significant association of BA.2.86 lineages with increased risk of disease progression) only in contexts where the true number of prior infections among cases infected with BA.2.86 lineages who avoided the need for emergency department or inpatient care was 6–27 times higher than that observed (Fig. 4). Under such conditions, the mean number of prior infections among cases infected with BA.2.86 lineages spanned 6.6–16.7, and exceeded the mean number of prior infections among cases infected with non-BA.2.86 lineages by a factor of 3.5–4.2, translating to 5.0–11.8 total unascertained infections. Cases infected with BA.2.86 lineages who avoided emergency department presentation or hospital admission would have experienced 5.6–11.5 more infections, on average, than those who experienced these outcomes.

While cases' true number of unascertained prior infections cannot be known, these figures likely exceed plausible levels based on estimates of SARS-CoV-2 reporting completeness during various phases of the pandemic[22,23]. Among all cases in the study population, only 0.6, 0.06, and 0.0005% were observed to have experienced 3, 4, or 5 prior infections throughout follow-up. Furthermore, the mean numbers of documented prior infections were only 1.2-fold higher among cases infected with BA.2.86 lineages than among cases infected with non-BA.2.86 lineages (0.6 and 0.5 documented prior infections on average, respectively, in the two case populations).

We next tested the impact of these alternative infection histories on the independent association of COVID-19 vaccination with the likelihood that cases were infected with BA.2.86 or non-BA.2.86 lineages. Allowing for greater numbers of prior infections according to the same formulation used in our analyses of clinical severity modestly strengthened the estimated association of prior vaccination with the detection of BA.2.86 lineages (Figs. S1, S2). Extending the analysis framework to allow for the differential history of unobserved prior infections associated with individuals' vaccination status, the

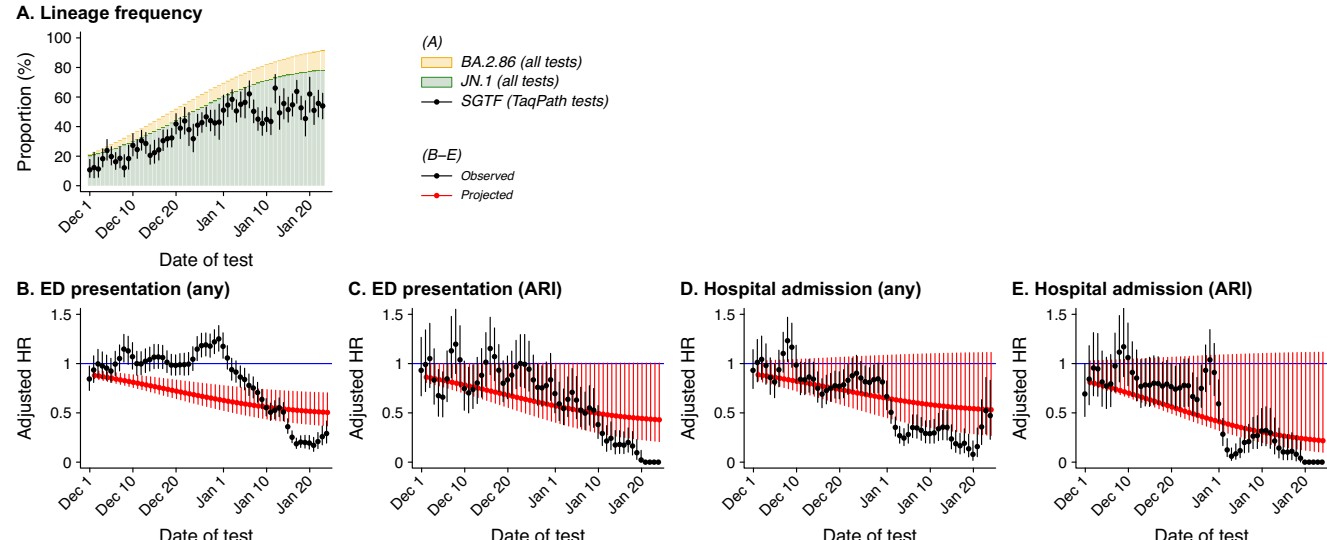

**Fig. 3 | Observed and projected changes in risk of progression.** Panels illustrate **A** the daily proportion of outpatient tests exhibiting S-gene target failure (BA.2.86-derived lineages) among all tested within the primary analytic cohort; **B** estimates of the observed day-specific adjusted hazard ratio of emergency department (ED) presentation due to any cause (black), as well as projected estimates of the day-specific adjusted hazard ratio of ED presentation due to any cause resulting only from changes in lineage composition among outpatient-diagnosed cases (defined as $\widehat{aHR}_t$ in the Methods; red); **C** for the outcome of hospital admission due to any cause, corresponding estimates of the observed day-specific hazard ratios and projected adjusted hazard ratios based only on changes in lineage composition (black and red, respectively); **D** for the outcome of ED presentations associated with acute respiratory infection (ARI) diagnoses, corresponding estimates of the observed day-specific hazard ratios and projected adjusted hazard ratios based only on changes in lineage composition (black and red, respectively); and **E** for the outcome of hospital admissions associated with acute respiratory infection (ARI) diagnoses, corresponding estimates of the observed day-specific hazard ratios and projected adjusted hazard ratios based only on changes in lineage composition (black and red, respectively). Analyses include data from 46,067 eligible individuals throughout the study period. For all panels (**A**–**E**), points indicate maximum likelihood estimates, with surrounding lines delineating 95% confidence intervals; we generated estimates via Cox proportional hazards models. Source data to replicate the figure are provided as a Source Data file (fig3_source.xlsx).

estimated association of vaccination with detection of BA.2.86 lineages was strengthened under scenarios where unobserved prior infections were more numerous among individuals who received fewer vaccine doses or had not received Omicron-adapted vaccine doses (Fig. S3). This scenario is consistent with empirical relationships within the study population of prior vaccination and prior documented infections (Table S8). In contrast, the estimated association was attenuated when considering scenarios where unobserved prior infections were more numerous among individuals who received greater numbers of vaccine doses or who received Omicron-adapted vaccine doses—a pattern inconsistent with the observed relationship between vaccination and prior documented infections.

## Discussion

Our study identifies escape of immune responses derived from prior COVID-19 vaccination or SARS-CoV-2 infection as a likely factor in the emergence of BA.2.86/JN.1 lineage during the period from December, 2023 to January, 2024. Cases infected with BA.2.86 lineages had 38, 51, and 60% higher adjusted odds of having received 5, 6, and ≥7 COVID-19 vaccine doses in comparison to cases infected with co-circulating lineages, predominantly descending from XBB. Although under-detection of prior infections limited our ability to compare infection history among cases infected with BA.2.86 lineages or non-BA.2.86 lineages, cases infected with BA.2.86 lineages had at least 8, 13, and 30% higher adjusted odds of having experienced 1, 2, or ≥3 prior documented SARS-CoV-2 infections. Overcoming concerns about statistical power within these primary analyses, our findings were reflected in period-based analyses in which prior vaccination and infection were each more strongly associated with diagnosis during phases when the JN.1 lineage accounted for a greater share of new SARS-CoV-2 infections. These findings suggest that immune responses resulting from prior vaccination or infection may have conferred greater protection against infection with XBB-derived lineages than BA.2.86 lineages.

Our findings that cases infected with BA.2.86 lineages had greater odds of prior infection during periods when XBB lineages were dominant in circulation, and greater odds of having received Omicron-targeted COVID-19 vaccine doses, are consistent with previous evidence of immune evasion by BA.2.86 and the JN.1 lineage, in particular. Sera from individuals previously infected with XBB.1.5 lineages have shown greater capacity to neutralize EG.5 and other XBB "FLip" lineages in comparison to BA.2.86[24]. Moreover, sera from XBB.1.5-targeted monovalent vaccine recipients exhibited superior neutralization of the XBB-derived EG5.1 and HK.3 lineages in comparison to sera from BA.4/BA.5-targeted bivalent vaccine recipients[25]; in contrast, sera from recipients of XBB.1.5-targeted and BA.4./BA.5-targeted vaccines had weak, non-differential capacity to neutralize the JN.1 lineage. On the surface, greater frequency of prior infection during the XBB-dominant period among cases infected with BA.2.86 lineages appears consistent with a scenario in which ancestral XBB lineages induced specific cross-protection against descendant lineages such as EG.5, HK.3, HV.1, JD.1, and JG.3. However, we also identified that cases infected with BA.2.86 lineages were more likely to have been infected during the BA.2-dominant period. Thus, differences in the ability of BA.2.86 lineages and non-BA.2.86 lineages to evade immune responses associated with prior XBB infection may be comparable to differences in their ability to evade immune responses associated with ancestral BA.2 lineages. Misclassification of individuals' prior infection status due to the occurrence of infections that were not ascertained through clinical testing is expected to attenuate the strength of the estimated association of infecting lineage with prior infection. Thus, our findings should be interpreted as lower-bound estimates of differences in the extent to which BA.2.86 lineages and co-circulating lineages evade immune responses associated with prior naturally-acquired infection. Unfortunately, changes in testing efforts over time impede direct comparison of effect size estimates for associations

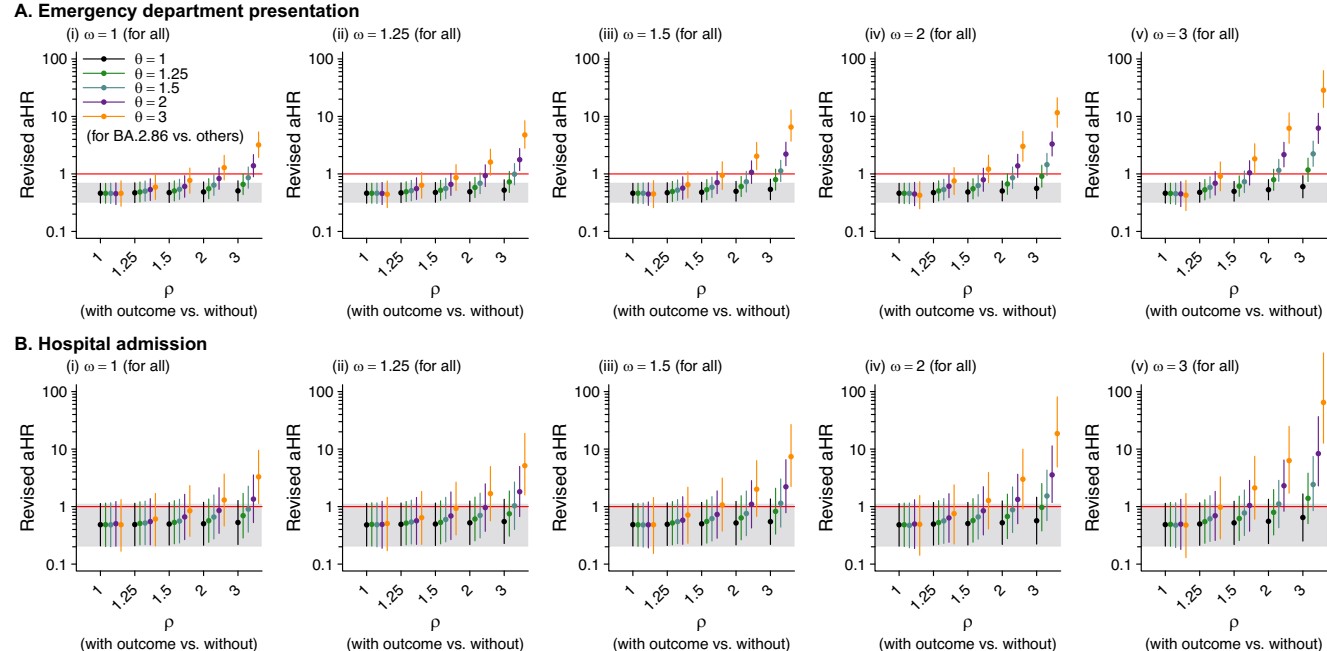

**Fig. 4 | Sensitivity analyses addressing the association of BA.2.86 lineage detection with risk of emergency department presentation and hospital admission in the presence of differential misclassification of prior infection according to infecting lineage and clinical outcome.** We illustrate estimates of the adjusted hazard ratio of progression to **A** emergency department presentation and **B** hospital admission under analyses imputing "full" infection histories for cases under the assumption of differential misclassification of prior infection status. We consider multipliers of 1, 1.25, 1.5, 2, and 3 for the ratio of true to observed infections, first non-differentially among all cases (ω), and for the relative ratio of true to observed infections comparing cases who evaded the indicated outcome versus those who experienced it (ρ) and comparing cases infected with BA.2.86-derived lineages to those infected with non-BA.2.86 lineages (θ). Analyses include data from 46,067 eligible individuals throughout the study period. For all panels (**A**, **B**), points indicate maximum likelihood estimates, with surrounding lines delineating 95% confidence intervals; we generated estimates via Cox proportional hazards models. Gray bands illustrate 95% confidence intervals for estimates from the primary analysis (Table 3). Source data to replicate the figure are provided as a Source Data file (fig4_source.xlsx).

of infecting lineage with documented infection during differing periods of the COVID-19 pandemic.

In an early study including 679 COVID-19 cases, XBB.1.5-targeted vaccination showed modestly weaker effectiveness against infection with BA.2.86 lineages in comparison to XBB-derived lineages (49 vs. 60%[19]), consistent with findings from a cohort study in which recipients of XBB.1.5-targeted monovalent vaccine had lower odds of infection with XBB-derived lineages in comparison to BA.2.86 lineages[26]. In a larger study of 6,551 adults with clinically-attended COVID-19, XBB.1.5-targeted monovalent vaccine conferred weaker effectiveness against progression from outpatient to emergency department or inpatient levels of care delivery among cases infected with BA.2.86 lineages than among cases infected with XBB-derived lineages (38 vs. 72%[27];). While these findings suggest the strength of protection afforded by XBB.1.5-targeted monovalent vaccines against BA.2.86 lineages and other lineages may be differential, it is important to note that all available evidence indicates XBB.1.5-targeted monovalent vaccines remain protective against BA.2.86 lineages including JN.1[19,26–28]. Thus, although individuals with specific immune protection against XBB-derived lineages may have provided a niche facilitating the expansion of BA.2.86 lineages, including JN.1, XBB.1.5-targeted monovalent vaccines remained a clinically relevant strategy for mitigating COVID-19 burden throughout the study period. Differences in the severity of infections captured across studies of the protective effectiveness of XBB.1.5-targeted monovalent vaccines against BA.2.86 lineages and other lineages are important to consider when comparing effect size estimates across studies.

Several findings from our study suggested attenuation of disease severity in BA.2.86 lineages. First, we estimated that cases diagnosed in outpatient settings who were infected with BA.2.86 lineages had a 54% lower risk of emergency department presentation than cases who were infected with non-BA.2.86 lineages. We obtained similar point estimates of differences in risk of progression to hospital admission, and greater point estimates of differences in risk for outcomes associated with ARI diagnoses. However, statistical power was constrained by the limited number of cases experiencing progression to these outcomes. Period-based analyses revealed continuous reductions in cases' risk of ARI-associated emergency department presentation and hospital admission throughout the period of JN.1 expansion, although changes over time in cases' risk of progression exceeded expectations based on lineage replacement alone. This may reflect differences over time in the clinical threshold at which cases sought testing, or other unmeasured differences over time in the characteristics of cases becoming infected; alternatively, the distribution of BA.2.86 sublineages identified among the primary analytic cohort may have differed from the distribution within the full case population, particularly by the end of the study period. While our inability to accurately classify cases' infection history may limit the interpretation of our findings, sensitivity analyses identified that implausible numbers of unobserved infections would be needed among cases infected with BA.2.86 lineages to reverse the direction of observed associations between infecting lineage and disease progression (mean 6.6–16.7 prior infections). No epidemiologic evidence supports a scenario in which appreciable numbers of individuals would have experienced this many SARS-CoV-2 infections prior to the study period. Further, the lack of meaningful differences in demographics, clinical characteristics, or healthcare-seeking behavior prior to diagnosis among cases infected with BA.2.86 lineages and non-BA.2.86 lineages make it unlikely that the proportion of prior infections that were ascertained should differ so appreciably between these groups.

Several additional limitations should be considered. First, as our study period coincided with multiple holidays, clinical thresholds at

which individuals sought SARS-CoV-2 testing and presented for subsequent care may have varied over time. While this poses limitations for period-based analyses, matching cases on their week of testing is likely to have alleviated the resulting bias in primary analyses. Moreover, KPSC maintained consistent criteria for hospital admission throughout the study period, and did not experience surges in severe COVID-19 cases that would necessitate tightening of such criteria. Second, our sample was identified through outpatient clinical testing, and may represent a more severe spectrum of all infections than what we would expect to identify through active, prospective testing of asymptomatic as well as symptomatic individuals. Thus, rates of progression to each clinical outcome should not be generalized to all infections. Third, because only a small proportion of samples were submitted for sequencing, we cannot distinguish clinical outcomes and characteristics of cases infected with JN.1 or other BA.2.86 lineages. Fourth, restricting our study to individuals with confirmed SARS-CoV-2 infection represents an instance of conditioning on a post-exposure variable for analyses of vaccination or prior infection[29]. Regardless, our case-only approach offered the advantage of selecting on healthcare-seeking behavior among cases regardless of their infecting lineage, which may otherwise represent a key source of bias in comparisons of individuals with differing histories of vaccination and prior infection[30]. Relatedly, our study framework comparing is observational in nature, and may be subject to unmeasured sources of confounding.

Risk of adverse clinical outcomes among cases diagnosed with SARS-CoV-2 infection in outpatient settings was low in our study population relative to those infected during earlier phases of the COVID-19 pandemic. Despite evidence from our study that BA.2.86 lineages may partially evade immunity acquired through prior vaccination (including with XBB.1.5-targeted boosters) and infection, and external evidence of extensive community transmission of these lineages[8], rates of COVID-19 hospital admissions and mortality did not match burden experienced during expansion of the Omicron BA.1, BA.4/BA.5, and XBB.1.5 lineages. Continued monitoring of the risk of these clinical outcomes, as well as vaccine effectiveness, remains important to inform response strategies to novel SARS-CoV-2 lineages, including the need for reformulation of booster doses.

## Methods

### Study population

Our study included all individuals who received positive molecular tests for SARS-CoV-2 infection between 1 November, 2023 and 30 January, 2024, who had no clinical record of receiving any positive SARS-CoV-2 test or COVID-19 diagnosis within 90 days before their first test ("index test") within this period, and who were enrolled in KPSC health plans for ≥1 year before their index test (allowing for lapses in membership of up to 45 days). We restricted the primary analytic cohort to individuals tested between 1 December, 2023 and 30 January, 2024 whose index tests were processed using Thermo Fisher TaqPath COVID-19 Combo Kit assays, for whom SGTF readout was available. Cases belonging to the primary analytic cohort were predominantly tested in outpatient-serving facilities without in-house laboratory facilities, which relied on regional centers using the Thermo Fisher TaqPath COVID-19 Combo Kit for specimen processing. Cases tested in hospital-based emergency departments had specimens processed in hospital laboratories via devices without SGTF readout.

### Exposures

We defined prior vaccine doses as those received >14 days before individuals' index test and distinguished doses received as monovalent wild-type (Wuhan-Hu-1) vaccines, bivalent BA.4/BA.5/Wuhan-Hu-1 vaccines, and monovalent XBB.1.5 vaccines. We defined prior infections as laboratory-confirmed SARS-CoV-2 diagnoses without any positive SARS-CoV-2 test result or COVID-19 diagnosis within the

preceding 90 days. We accounted for nirmatrelvir-ritonavir receipt as a time-varying covariate for all dispenses initiated within 5 days after the index test date; for dispenses initiated after the index date, individuals' exposure status was permitted to change beginning on the dispense date. Additional characteristics obtained from patient electronic health records and accompanying demographic metadata included cases' age (categorized in 10-year increments for all analyses), sex, race/ethnicity (categorized as White non-Hispanic, Black non-Hispanic, Hispanic of any race, Asian, Pacific Islander, or other/mixed/unknown race/ethnicity), body mass index (categorized as underweight, normal weight, overweight, or obese if measured in the preceding year), history of cigarette smoking (current, former, or never smokers), prior-year healthcare utilization (categorized across outpatient, emergency department, and inpatient settings as presented in Table 1), Charlson comorbidity index (0, 1–2, 3–5, or ≥6), and median household income within their census tract (categorized as presented in Table 1).

### Outcomes

Within the primary analytic cohort, we considered cases with positive detection ($c_T < 37$) of the SARS-CoV-2 S, N, and orf1a/b genes to be infected with non-BA.2.86 lineages. We considered cases with positive detection of N and orf1a/b genes but no detection ($c_T \geq 37$) of the S gene (SGTF) to be infected with BA.2.86 lineages. We followed cases for the following outcomes within the specified time range from the index test: any emergency department presentation within 14 days; ARI-associated emergency department presentation within 14 days; any hospital admission within 28 days; ARI-associated hospital admission within 28 days; ICU admission within 60 days; initiation of mechanical ventilation within 60 days; and death within 60 days. Due to the low frequency of severe outcomes, we defined a composite outcome of ICU admission, initiation of mechanical ventilation, or death within 60 days. For each case, we defined analysis periods ending at the occurrence of any study outcome or censoring due to disenrollment; a new observation window was initiated at the point of nirmatrelvir-ritonavir dispense if treatment preceded any study outcome. For analyses of ARI-associated emergency department presentations and hospital admissions, we also censored observations at the occurrence of the outcome without any accompanying ARI diagnosis code.

### Missing data

Among 68,281 cases, 14,563 (21%) had missing entries for at least one analytic variable. Missing value frequencies were as follows: 9630 (14%) for body mass index, 8535 (12.5%) for cigarette smoking, 4129 (6%) for census tract median household income, and 1102 (2%) for age. We populated five complete pseudo-datasets via multiple imputation using the Amelia package[31]. For all analyses, we pooled results from replications across each pseudo-dataset. We verified that results pooled across imputations were consistent with those obtained from complete-case analysis (excluding cases with missing data) and models fitted to the individual imputed datasets (Tables S9, S10).

### Comparison of prior vaccination and infection

For members of the primary analytic cohort, we fit conditional logistic regression models defining infection with BA.2.86 lineages or non-BA.2.86 lineages as the outcome variable and defining matching strata for their week of testing. Models controlled for the following covariates based on the expectation that they could confound the relationship between immune history and infecting lineage, by predicting both prior vaccination or infection and individuals' relative likelihood of being exposed to and acquiring infection with BA.2.86 lineages or non-BA.2.86 lineages: age, sex, race/ethnicity, body mass index, history of cigarette smoking, prior-year healthcare utilization across all settings, Charlson comorbidity index, and median household income within cases' census tract. We measured healthcare utilization via three

variables. The first enumerated total outpatient care encounters for each individual in the year preceding their test. The second indicated whether the individual had ever received care in an emergency department setting in the year preceding their test, and the third indicated whether the individual had ever been admitted in an inpatient setting in the year preceding their test. In total, analyses included 13 covariates, as listed in Table 1. Individuals were excluded from analyses if they received any vaccine dose within ≤14 days of their index test.

We also report results of analyses which distinguished receipt of wild-type (Wuhan-Hu-1) vaccine doses, BA.4/BA.5-targeted bivalent vaccine doses, and XBB.1.5-targeted vaccine doses, as well as counts of vaccine doses received and the timing of receipt of the most recent vaccine dose (<3 months, 3–6 months, or >6 months). Analyses distinguishing prior infection by periods when distinct SARS-CoV-2 variants were dominant in circulation (Table S4) defined no documentation of prior SARS-CoV-2 infection as the reference exposure.

For period-based analyses including all outpatient-diagnosed cases without restriction on test assay, we fit separate logistic regression models defining infection during each of the periods of 1–15 December, 2023, 16–31 December, 2023, 1–15 January, 2024, or 16–30 January, 2024 as outcomes ("1"), with infection during the period of 1–30 November, 2023 defined as the control outcome ("0"). Models again included the number of vaccine doses cases had received and the number of recorded infections preceding cases' index test. Cases who received vaccination on or after 1 November, 2023 were excluded to enable comparisons of exposures that all cases were eligible to encounter. Consistent with primary analyses, models controlled for age, sex, race/ethnicity, body mass index, history of cigarette smoking, prior-year healthcare utilization across all settings, Charlson comorbidity index, and median household income within cases' census tract. No adjustment for calendar time was included due to our specification of infection timing as the outcome variable.

**Comparison of clinical outcomes**
Within the primary analytic cohort, we fit Cox proportional hazards models for each outcome defining matching strata on cases' week of testing. We defined infection with BA.2.86 lineages or non-BA.2.86 lineages as the exposure of interest; outcomes were progression to emergency department presentation (associated with any diagnosis or with ARI diagnosis) within 14 days after testing, progression to hospital admission (associated with any diagnosis or ARI diagnosis) within 28 days after testing, progression to severe illness (defined as intensive care unit admission or initiation of mechanical ventilation) within 60 days after diagnosis, or death within 60 days after diagnosis. We recorded individuals' outcome as progression if they experienced a higher-severity prior to the studied outcome (e.g., hospital admission without preceding emergency department presentation, or death without preceding hospital admission). For analyses of ARI-associated outcomes, we censored observations at individuals' first ED presentation or hospital admission if these events occurred without ARI diagnoses. Models adjusted for the same covariates listed above for conditional logistic regression analyses, with the addition of a time-varying covariate for receipt of nirmatrelvir-ritonavir. We verified the proportional hazards assumption by visual inspection of Kaplan–Meier plots (Fig. S4) and by testing for non-zero slopes (with respect to time) of Schoenfeld residuals from fitted models[32]. For both lineage-based and period-based analyses of all study outcomes, this test yielded two-sided $p$ values greater than 0.1 (Table 3 and Fig. 2), providing no evidence of a violation.

For period-based analyses, including outpatient-diagnosed cases without restriction on test assay, we defined infection during the periods of 1–15 December, 2023, 16–31 December, 2023, 1–15 January, 2024, or 16–30 January, 2024 as the exposures of interest; the

reference period was 1–30 November, 2023, when BA.2.86 lineages did not circulate prominently. We controlled for the same covariates as those included in primary analyses, with no adjustment for calendar time. Again, cases were excluded if they received vaccination after 1 November, 2023.

To distinguish the role of both secular (time-varying) factors and infecting lineage in contributing to the reduced incidence of severe disease outcomes over the course of the study period, we also compared empirical estimates of day-specific hazard ratios for cases' risk of progression to each outcome to projections of day-specific hazard ratios based only on changes in SARS-CoV-2 lineage composition. Empirical estimates were fitted via Cox proportional hazards that included day-specific intercepts for each day from 1 December, 2023 through 30 January, 2024 (measured relative to risk for cases diagnosed between 1 and 30 November, 2023), and controlled for the same factors listed above. We projected corresponding day-specific estimates of the hazard ratio of progression due only to changes in SARS-CoV-2 lineage composition, $\widehat{\text{aHR}}_t$, via the formula

$$\widehat{\text{aHR}}_t^* = \text{aHR}_{\text{SGTF}}\pi_t + (1 - \pi_t) \quad (1)$$

where $\text{aHR}_{\text{SGTF}}$ indicated the adjusted hazard ratio of progression to the outcome of interest comparing cases infected with BA.2.86 lineages to cases infected with non-BA.2.86 lineages (as estimated in the primary analytic cohort), and $\pi_t$ indicated the proportion of cases infected at time $t$ with BA.2.86 lineages among all cases for whom sequencing results were available. Our derivation of $\widehat{\text{aHR}}_t^*$ proceeds as follows. Consider that $h(\text{BA.2.86})$ and $h(\text{Other})$ represent the (adjusted) hazards of progression for individuals infected with BA.2.86 lineages and other lineages, respectively, such that $\text{aHR}_{\text{SGTF}} = h(\text{BA.2.86})/h(\text{Other})$. Consider further that $h(t)$ represents the (adjusted) hazard of progression for cases infected with any lineage who are diagnosed at time $t$. Under a scenario where no factors besides infecting lineage influence individuals' risk of disease progression, $\widehat{\text{aHR}}_t$ measures the ratio $h(t)/h(\text{Other})$. We derive $h(t)$ according to

$$h(t) = h(\text{BA.2.86}) \Pr[\text{infection with BA.2.86}|t] + h(\text{Other})(1 - \Pr[\text{infection with BA.2.86}|t] \quad (2)$$

which is equal to $h(\text{BA.2.86})\pi_t + h(\text{Other})(1 - \pi_t)$, by definition. Dividing by $h(\text{Other})$ yields

$$\widehat{\text{aHR}}_t^* = \frac{h(t)}{h(\text{Other})} = \frac{h(\text{BA.2.86})\pi_t + h(\text{Other})(1 - \pi_t)}{h(\text{Other})} = \frac{h(\text{BA.2.86})}{h(\text{Other})}\pi_t + (1 - \pi_t) = \text{aHR}_{\text{SGTF}}\pi_t + (1 - \pi_t) \quad (3)$$

We generated day-specific estimates of $\pi_t$ by fitting a regression model to data representing weekly proportions of sequences found to represent BA.2.86 lineages; we defined polynomial transformations of calendar time as the independent variables, finding that a fifth-degree polynomial yielded the lowest value of the Bayesian information criterion (Fig. S5). We overlay our projected $\widehat{\text{aHR}}_t$ estimates with empirical day-specific adjusted hazard ratio estimates in Fig. 3.

**Sensitivity analyses**
To assess the sensitivity of our estimates to differential misclassification (undercounting) of prior infections, particularly among cases infected with BA.2.86 lineages and those who evaded severe clinical outcomes, we also conducted analyses imputing alternative infection histories among cases. Defining the prior number of infections for case $i$ within the primary analytic cohort as a Poisson random variable with the underlying rate $\lambda_i$, we sampled alternative infection histories $X_i$ as

Poisson random variables according to

$$X_i \sim \mathrm{Pois}\big(\omega\big[1 - \mathbb{I}(\mathrm{SGTF}_i)(1-\theta)\big]\big[1 - Y_i(1-\rho)\big]\lambda_i\big) \quad (4)$$

Here $\omega$ provided a multiplier conveying the minimum ratio of true to documented infections among all cases; if under-detection was considered differential for cases according to infecting lineage or a clinical outcome of interest, $\omega$ conveyed the ratio of true to documented infections among cases infected with non-BA.2.86 lineages ($\mathbb{I}(\mathrm{SGTF}_i) = 0$) who experienced the outcome ($Y_i = 1$). The parameters $\theta$ and $\rho$, respectively, represented the relative ratio of true to documented infections among cases infected with BA.2.86 lineages (relative to cases infected with non-BA.2.86 lineages) and the relative ratio of true to documented infections among cases who evaded each clinical outcome (relative to those who experienced the outcome). We considered values of 1, 1.25, 1.5, 2, and 3 for $\omega$, $\rho$, and $\theta$, so that the corrected rate parameter could be up to 27-fold higher than that observed for cases infected with BA.2.86 lineages who evaded each clinical outcome.

We estimated $\lambda_i$ via Poisson regression models defining cases' number of prior infections as the outcome variable and including all other measured covariates as predictors. For each parameterization of $\{\omega, \rho, \theta\}$, we drew 10 vectors of case infection histories $\hat{X}_i$ and fit Cox proportional hazards models according to the specifications of the primary analysis.

For sensitivity analyses addressing the impact of prior unobserved infections on measured associations of vaccination with infecting lineage, we extended the above formulation to further account for scenarios where individuals' vaccination status was associated with the number of infections they had experienced. We first considered a scenario where individuals who had received fewer vaccine doses, or who had not received Omicron-adapted vaccines, had experienced greater numbers of prior infections; this circumstance was consistent with the observed association of prior documented infection with individuals' vaccination status (Table S8) and with a hypothesis that prior vaccination was associated with protection against such infections or with other behavioral characteristics that reduced individuals' risk of infection[33]. Here, we defined

$$X_i \sim \mathrm{Pois}\big(\omega\big[1 - \mathbb{I}(\mathrm{SGTF}_i)(1-\theta)\big]\big[1 - (1-Z_i)(1-\sigma)\big]\lambda_i\big), \quad (5)$$

taking $Z_i$ as an indicator for vaccine status of interest (receipt of XBB.1.5 monovalent vaccine, BA.4/BA.5 bivalent vaccine, or receipt of a specified number of vaccine doses), and defining $\sigma$ equal to 1, 1.5, or 2 to convey the relative risk of infection for individuals with $Z_i = 0$. To consider an alternative scenario where prior documented infections were instead more common among individuals with $Z_i = 1$, we defined

$$X_i \sim \mathrm{Pois}\big(\omega\big[1 - \mathbb{I}(\mathrm{SGTF}_i)(1-\theta)\big]\big[1 - Z_i(1-\sigma)\big]\lambda_i\big). \quad (6)$$

## Software
We conducted analyses using R software (R Foundation for Statistical Computing, Vienna, Austria). We used the Amelia package[31] for multiple imputation and fit conditional logistic regression models and Cox proportional hazards models using the survival package[34].

## Ethics
This study was reviewed and approved by Kaiser Permanente Southern California Institutional Review Board and the US Centers for Disease Control and Prevention (CDC) and was conducted consistent with applicable federal law and CDC policy (45 CFR part 46, 21 CFR part 56; 42 USC Sect. 241 (d); 5 USC Sect. 552a; 44 USC Sect. 3501 et seq.).

## Reporting summary
Further information on research design is available in the Nature Portfolio Reporting Summary linked to this article.

## Data availability
Individual-level testing and clinical outcomes data reported in this study are not publicly shared due to privacy protections for patient electronic health records. Individuals wishing to access disaggregated data, including data reported in this study, should submit requests for access to sara.y.tartof@kp.org. Requests will receive a response within 14 days. De-identified data (including, as applicable, participant data and relevant data dictionaries) will be shared upon approval of analysis proposals with signed data-access agreements in place. Source data to replicate figures are provided with this paper. Source data are provided with this paper.

## Code availability
Analysis code is available from GitHub[35].

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

## Acknowledgements

This study was funded by the US Centers for Disease Control and Prevention (75D30123-C-18129 SYT) and the US National Institutes of Health (RO1-AI148336 to J.A.L.). The findings and conclusions in this report are those of the authors and do not necessarily represent the official position of the Centers for Disease Control and Prevention.

## Author contributions

J.A.L., R.L.-G., L.R.F., and S.Y.T. contributed to the study concept and design. V.H. and S.Y.T. led the acquisition of data. J.A.L. led the statistical analysis of data. J.A.L., P.M., D.M., B.K.A., B.J.L., R.L.-G., L.R.F., M.L., and S.Y.T. led the interpretation of data. J.A.L. drafted the manuscript, and all authors critically revised the manuscript for important intellectual content. S.Y.T. obtained funding and provided supervision.

## Competing interests

J.A.L. has received research grants paid directly to his institution and consulting honoraria unrelated to this study from Pfizer. S.Y.T. has received research grants paid directly to her institution unrelated to this study from Pfizer. The remaining authors declare no competing interests.
