## [Peer Review File · Nature Communications]

Immune escape and attenuated severity associated with the SARS-CoV-2 BA.2.86/JN.1 lineageREVIEWER COMMENTS

Reviewer #1 (Remarks to the Author):

This is a relevant and well-conducted study showing (some) immune escape and lower severity of infections caused by the BA2.86/JN.1 variant in comparison with infections caused by the XBB variant. The authors use individual level data on the infecting strain and try to confirm their findings with more data but comparing time periods with different distributions of JN.1 vs XBB variants. They also provide an interesting additional analysis to try disentangling variant effects and calendar time effects. And a sensitivity analysis on misclassification of prior infection.

Comments:

1. The estimates for XBB.1.5 vaccination and prior infection are very small (1.14 and 1.09, respectively) and also smaller than previous studies (that found ~ 1.5 for XBB.1.5 vaccination). Although the estimates are statistically significant, do the authors think these differences are clinically relevant? They could elaborate on this in the discussion.
2. A sensitivity analysis on misclassification of prior infection is done for the severity analyses. Was this also done for the immune escape analyses? Why not? Also there this could be relevant.
3. It would be helpful to give more information on the vaccination recommendation/program during the study period. Depending on that recommendation/program it could be more relevant to look specifically at XBB.1.5 vaccination rather than number of doses.
4. To be able to better compare the individual-data level analysis with the calendar time analysis, a table with estimates from both methods would be helpful.
5. The part on nirmatrelvir-ritonavir receipt as time-varying exposure in the methods section is not completely clear. The term time-varying covariate which is used later may be more applicable.

Reviewer #2 (Remarks to the Author):

Please refer to my attached report.

Review report of NCOMMS-24-27655: “Immune escape and attenuated severity associated with the SARS-CoV-2 BA.2.86/JN.1 lineage”

Anonymous Reviewer

June 2024

The authors used a moderately-sized retrospective cohort study to investigate immune escape and infection severity of SARS-CoV-2 BA.2.86/JN.1 lineage. The patient cohort is extracted from the Kaiser Permanente Southern California healthcare system during December 2023 to January 2024. Some statistical analyses were performed; the authors identified immune escape given prior vaccination and/or infection, and found attenuated severity in clinical outcomes after infection with SARS-CoV-2 BA.2.86/JN.1 lineage.

Virology or epidemiology is not my expertise, so I will only comment on the methods, data analysis and interpretation.

1. About missing data: given that 21% of records have at least one variable missing, the authors should consider a complete-case analysis and compare the results with multiple imputation. And it would be useful to also show the variation of results across different imputations to check consistency.
2. In the conditional logistic regression models, what variables are being captured for “prior-year healthcare utilization across all settings”? It seems that there are a lot of covariates included in the regression model – have the authors considered regularization approaches to do variable selection and/or avoid overfitting? (It would be helpful to mention how many variables are included to give an idea about the severity of potential over-parameterization.)
3. For the Cox proportional hazards model, the authors should at least check the Kaplan Meier curves to validate the proportional hazard assumption.
4. The projected day-specific estimates of the hazard ratio of progression (to each clinical outcome) do not make much sense. The interpretation of a “hazard ratio” is the ratio of hazards for an outcome under two conditions for one individual, with everything else held constant. It is unclear how to interpret the \widehat{aHR}^*_t as it is somehow modulated with population-level proportions of those two conditions under comparison – also, this rather awkward definition as a weighted average between aHR and 1 in the methods seems incorrect, because **hazard ratios are not additive**, but rather, the hazards are.

If the authors must mix population-level proportions and hazard ratios, then we can think of the regular aHR estimate as the hazard ratio of progression when BA.2.86 and non-BA.2.86 lineage in the population are 1:1 in prevalence (though this still doesn’t make much sense). So the population-proportion modulated version could perhaps be defined as:

$$\widehat{aHR}^*_t = aHR \frac{\pi_t}{1 - \pi_t}.$$

Again, I’m not sure how this “projected HR” is informative. As shown in Figure 3, the projected day-specific HR estimates for each outcome simply decline over time (panels B-E) as BA.2.86 lineage frequency goes up (panel A), and it’s rather hard to explain why such projection is so different (in both scale and trend) from the empirical estimates obtained from observed data.

5. For the “comparison of clinical outcomes” analysis, how did the authors deal with the time lag between infection and a clinical outcome? In the description of methods, it looks like the authors are simply relating outcome at time t with infection at time t as well?

6. Figure 2: the authors should have the y-axis in each panel on the same scale for easier comparison. Right now the y-axes are all over the place, even with a weird breakpoint in panel C (that's bad practice for visualization).
7. Figures 2 and 3: it would be better to have the panels aligned horizontally rather than vertically.

REVIEWER COMMENTS

Reviewer #1 (Remarks to the Author):

This is a relevant and well-conducted study showing (some) immune escape and lower severity of infections caused by the BA.2.86/JN.1 variant in comparison with infections caused by the XBB variant. The authors use individual level data on the infecting strain and try to confirm their findings with more data but comparing time periods with different distributions of JN.1 vs XBB variants. They also provide an interesting additional analysis to try disentangling variant effects and calendar time effects. And a sensitivity analysis on misclassification of prior infection.

We thank the Reviewer for their assessment and for the constructive comments below, each of which we have addressed in this Revision.

Comments:

19, 26, 27

Link Gelles/Huiberts/Levy

1. The estimates for XBB.1.5 vaccination and prior infection are very small (1.14 and 1.09, respectively) and also smaller than previous studies (that found ~1.5 for XBB.1.5 vaccination). Although the estimates are statistically significant, do the authors think these differences are clinically relevant? They could elaborate on this in the discussion.

We thank the Reviewer for raising this point. We have extended the Discussion to further address the clinical significance of the finding (lines 239-245) as well as the comparison of our estimate to those coming from other studies (lines 245-247).

Briefly, we believe that although our finding helps to understand the evolution of the JN.1/BA.2.86 lineages from a scientific standpoint, the clinical significance is modest: available data suggest indicate the XBB-targeted vaccine was moderately effective against infection with the JN.1/BA.2.86 lineages. In terms of the other studies, estimates in the ~1.5 come from two prior studies in Denmark and the Netherlands, which we have cited in this revision. In contrast to our study, which used specimens obtained in clinical settings only, these studies included specimens from programs where individuals could self-test at home or at work, which likely led to capture of a less-severe spectrum of illness than what was captured in clinical settings within our study. Since reductions in vaccine effectiveness against novel variants tend to be greatest for less-severe outcomes (e.g. asymptomatic or mild infection) and protection is generally more robust for more-severe disease, we believe this design characteristic is most likely to account for differences in our estimates. In comparison to the other studies, our study also had access to a broader set of individual-level covariates, which we used to correct for potential confounding. A comparison of our adjusted and unadjusted findings shows this also contributed to reducing the point estimates.

2. A sensitivity analysis on misclassification of prior infection is done for the severity analyses. Was this also done for the immune escape analyses? Why not? Also there this could be relevant.

We have extended the sensitivity analysis to further explore how the scenarios considered, in terms of additional unobserved infections, could impact findings of our “immune escape” analysis. As the sensitivity analysis requires us to manipulate the number of prior infections differentially for individuals infected with BA.2.86 and non-BA.2.86 lineages, the inferential question that persists is the strength of the independent association of vaccination with individuals’ relative likelihood of infection with BA.2.86 or non-BA.2.86 lineages after accounting for the fact the number of unobserved prior infections could differ among better-vaccinated individuals (in terms of number or type of vaccines received) and less-vaccinated individuals.

We find that the independent association of vaccination with infecting lineage is strengthened after accounting for possible unobserved infection. Under the same parameterization as we used for the severity analyses, we obtained greater point estimates of effect sizes for the association of BA.2.86 lineage detection with receipt of XBB.1.5 monovalent vaccine, receipt of BA.4/BA.5 bivalent vaccine, and receipt of 5 or 6 COVID-19 vaccine doses. We have revised the r

3. It would be helpful to give more information on the vaccination recommendation/program during the study period. Depending on that recommendation/program it could be more relevant to look specifically at XBB.1.5 vaccination rather than number of doses.

We have revised the text to indicate clearly that the XBB-targeted vaccine was recommended during the study period (lines 59-60) and have revised our presentation of the results to first address differences according to receipt of this vaccine, rather than leading with the results based on number of doses (lines 60-68). Similarly, we have updated Table 2 to lead with the results based on receipt of the XBB-targeted vaccine rather than number of doses.

4. To be able to better compare the individual-data level analysis with the calendar time analysis, a table with estimates from both methods would be helpful.

We have added Table S5 presenting results from both approaches to enable side-by-side comparison.

5. The part on nirmatrelvir-ritonavir receipt as time-varying exposure in the methods section is not completely clear. The term time-varying covariate which is used later may be more applicable.

We have revised this language as suggested (line 318).

The authors used a moderately-sized retrospective cohort study to investigate immune escape and infection severity of SARS-CoV-2 BA.2.86/JN.1 lineage. The patient cohort is extracted from the Kaiser Permanente Southern California healthcare system during December 2023 to January 2024. Some statistical analyses were performed; the authors identified immune escape given prior vaccination and/or infection, and found attenuated severity in clinical outcomes after infection with SARS-CoV-2 BA.2.86/JN.1 lineage.

Virology or epidemiology is not my expertise, so I will only comment on the methods, data analysis and interpretation.

1. About missing data: given that 21% of records have at least one variable missing, the authors should consider a complete-case analysis and compare the results with multiple imputation. And it would be useful to also show the variation of results across different imputations to check consistency.

We have added two supplemental tables presenting results for complete-case analyses and for the individual imputed datasets (Table S8, Table S9; lines 345-346). Briefly, the results are highly consistent for complete case analyses and across all imputations, with both point estimates and confidence interval bounds varying only by a few hundredths (or less).

2. In the conditional logistic regression models, what variables are being captured for “prior-year healthcare utilization across all settings”? It seems that there are a lot of covariates included in the regression model – have the authors considered regularization approaches to do variable selection and/or avoid overfitting? (It would be helpful to mention how many variables are included to give an idea about the severity of potential over-parameterization.)

We have clarified that prior-year healthcare utilization is measured via three variables as specified in Table 1. These are (1) the number of outpatient encounters for any reason in the prior year; (2) whether individuals had any emergency room encounter in the prior year; and (3) whether individuals had any hospital admission in the prior year (lines 352-357). In total, the model includes 13 variables; accounting for all unique categories of each variable, there 38 unique coefficients estimated. Given this model includes 7,694 unique individuals (>200 observations/coefficient estimated), and all exposures have >50 persons per stratum, the sample size is adequate to support estimation.

More fundamentally, it is important to note that the fundamental objective of this analysis is inference rather than prediction. Whereas regularization and variable selection approaches offer value for prediction, the objective of our analysis is not to support out-of-sample prediction, nor are we dealing with high dimensionality. Covariates included in the model are included because they are expected to confound the relationship between the exposure of interest (vaccination or prior infection) and the outcome (infection with one variant or another). This is consistent with the causal framework described by Greenland, Pearl & Robins (*Epidemiology* 1999; PMID: 9888278); a nice description on the distinct objectives of DAG-based causal inference versus analyses that would employ variable selection/regularization is available in Heinze, Wallisch & Dunkler (*Biometrical Journal* 2018; PMID 29292533). We have clarified this objective and framework in the revised Methods section (lines 350-353).

To verify that the exposures which were a focus of this analysis are likewise identified to be of importance under regularization-based approaches, we include below a table of coefficient values for a logistic LASSO regression analysis as an alternative to the analysis presented in the manuscript. While estimates are not head-to-head comparable between the approaches (e.g. we have specified these as continuous variables and the LASSO applies shrinkage), we hope this helps to allay any concern that our primary analysis is reporting spurious associations.

Covariate	Coefficient
Number of vaccine doses (continuous)	0.087
Number of prior infections (continuous)	0.029
Charlson comorbidity index	-0.082
Age (continuous)	-0.020
Week of testing	0.227

3. For the Cox proportional hazards model, the authors should at least check the Kaplan Meier curves to validate the proportional hazard assumption.

We have added Figure S1 showing the Kaplan Meier curves for all outcomes; visually there is no evidence of a violation. More formally, we have verified the proportional hazards assumption by testing for non-zero slopes with respect to time in the Schoenfeld residuals from the fitted models (lines 380-383), which similarly identified no violation of the proportional hazards assumption. We have added this note to the caption to Table 3 and the caption to Figure 2.

4. The projected day-specific estimates of the hazard ratio of progression (to each clinical outcome) do not make much sense. The interpretation of a “hazard ratio” is the ratio of hazards for an outcome under two conditions for one individual, with everything else held constant. It is unclear how to interpret the \widehat{aHR}_t^* as it is somehow modulated with population-level proportions of those two conditions under comparison – also, this rather awkward definition as a weighted average between aHR and 1 in the methods seems incorrect, because **hazard ratios are not additive**, but rather, the hazards are.

If the authors must mix population-level proportions and hazard ratios, then we can think of the regular aHR estimate as the hazard ratio of progression when BA.2.86 and non-BA.2.86 lineage in the population are 1:1 in prevalence (though this still doesn’t make much sense). So the population proportion modulated version could perhaps be defined as:

$$\widehat{aHR}_t^* = aHR \frac{\pi_t}{1 - \pi_t}.$$

Again, I’m not sure how this “projected HR” is informative. As shown in Figure 3, the projected day-specific HR estimates for each outcome simply decline over time (panels B-E) as BA.2.86 lineage frequency goes up (panel A), and it’s rather hard to explain why such projection is so different (in both scale and trend) from the empirical estimates obtained from observed data.

We have sought to clarify the points raised above in the revised manuscript (lines 406-418).

First, have sought to be clear in defining the interpretation of the “regular” aHR estimate (for which we use the notation aHR_{SGTF}): this estimate compares the hazards of progression for an individual who is infected with the BA.2.86 lineage to a counterfactual scenario where the same individual was infected with a non-BA.2.86 lineage. This is distinct from the definition the Reviewer states above (“we can think of the regular aHR estimate as the hazard ratio of progression when BA.2.86 and non-BA.2.86 lineage in the population are 1:1 in prevalence”) as the parameter is independent of the composition of lineages in the population—whether the lineages are in 1:99, 1:1, or 99:1 frequency, aHR_{SGTF} is the same.

We have introduced the terms $h(\text{BA.2.86})$ and $h(\text{Other})$ as the adjusted hazards of progression for an individual infected with BA.2.86 or with other lineages, so that the “regular” aHR can be formulated as

$$aHR_{SGTF} = \frac{h(\text{BA.2.86})}{h(\text{Other})}.$$

Similarly, the daily hazard ratios plotted in Figure 3 define a period in which there is no evidence of BA.2.86 circulation as the referent (i.e., denominator analogous to $h(\text{Other})$ in the formulation above). We hope this clarifies any misunderstanding about a hazard ratio involving 1:1 prevalence of the circulating lineages, as this is neither an assumption nor a condition addressed by our analysis.

Under a scenario where there are no factors besides infecting lineage that cause differences over time in individuals’ hazard of progression, the hazard of progression for cases diagnosed at time t , which we denote $h(t)$, can be obtained as follows:

$$h(t) = h(\text{BA.2.86}) \Pr[\text{infected with BA.2.86}|t] + h(\text{Other})(1 - \Pr[\text{infected with BA.2.86}|t])$$

Since we define $\pi_t = \Pr[\text{infected with BA.2.86}|t]$, this is equivalent to

$$h(t) = h(\text{BA.2.86})\pi_t + h(\text{Other})(1 - \pi_t)$$

We define \widehat{aHR}_t^* as the relative hazard of progression for an individual infected at time t , when a mixture of BA.2.86 and non-BA.2.86 lineages were in circulation, compared to the same individual’s counterfactual hazard if they were infected with a non-BA.2.86 lineage (again, assuming that all differences in progression risk are accounted for by the lineage individuals are infected with). Thus,

$$\widehat{aHR}_t^* = \frac{h(t)}{h(\text{Other})} = \frac{h(\text{BA.2.86})\pi_t + h(\text{Other})(1 - \pi_t)}{h(\text{Other})} = \frac{h(\text{BA.2.86})}{h(\text{Other})}\pi_t + (1 - \pi_t).$$

Substituting for aHR_{SGTF} yields the equation we presented in the original submission, i.e.

$$\widehat{aHR}_t^* = aHR_{SGTF}\pi_t + (1 - \pi_t).$$

While we hope it is helpful to offer this clarification, we agree that this is neither the most important nor the most informative analysis within the paper; the \widehat{aHR}_t^* just provides a benchmark in Figure 3 showing that the reduction in risk of severe outcomes is bigger than expected, which might owe to additional factors (changes in healthcare seeking behavior over time or in the specific composition of non-BA.2.86 lineages). As the original submission may have placed more emphasis than warranted on this ancillary analysis, we have revised the text to address the point much concisely in two sentences (lines 154-158). Together with this clarification of the framework for how we derive \widehat{aHR}_t^* , we hope that rationale for and interpretation of this analysis are now clearer.

5. For the “comparison of clinical outcomes” analysis, how did the authors deal with the time lag between infection and a clinical outcome? In the description of methods, it looks like the authors are simply relating outcome at time t with infection at time t as well?

We have clarified that this is a time-to-event analysis: we follow cases over time from their index test in an outpatient setting for the specified clinical outcomes (emergency department presentation for any cause within 14 days; emergency department presentation with ARI diagnosis within 14 days; hospital admission for any cause within 28 days; and hospital admission with ARI diagnosis within 28 days). This point is now specified in the Results (lines 120-121) as well as the Methods (lines 367-374).

In other words, infection at time t is not compared to outcome at time t ; we follow for outcomes occurring after the infection was initially identified in an outpatient setting, as this is a survival analysis.

6. Figure 2: the authors should have the y-axis in each panel on the same scale for easier comparison. Right now the y-axes are all over the place, even with a weird breakpoint in panel C (that’s bad practice for visualization).

To facilitate comparisons of panels A and B we have aligned these figures horizontally (as suggested below) with the same y-axis. Panel C, however, presents estimates of a distinct parameter (hazard ratio rather than odds ratio) and concerns a different outcome (progression to emergency department or inpatient care in panel C, versus infecting lineage in panels A and B). Thus, comparison of estimates between panels A/B and panel C is not possible, and is not a priority for visualization. However, in agreement with the Reviewer’s comment about good practice, we have removed the axis break in panel C.

7. Figures 2 and 3: it would be better to have the panels aligned horizontally rather than vertically.

For Figure 2, we have aligned panels A and B horizontally as described above; we have also aligned panels of Figure 3 horizontally as suggested.

REVIEWERS' COMMENTS

Reviewer #1 (Remarks to the Author):

The authors have adequately addressed my comments.

Reviewer #2 (Remarks to the Author):

Thank you for addressing all my comments carefully and highlighting the edits in the revised manuscript. I appreciate the clarification you have made in the response too. I don't have any further comments.